# Concurrent and future risk of endometrial cancer in women with endometrial hyperplasia: A systematic review and meta-analysis

**Michelle T. Doherty**[1☉], **Omolara B. Sanni**[1☉], **Helen G. Coleman**[1,2], **Chris R. Cardwell**[1], **W. Glenn McCluggage**[3], **Declan Quinn**[4], **James Wylie**[4], **Úna C. McMenamin**[1]*

1 Cancer Epidemiology Research Group, Centre for Public Health, Queen's University Belfast, Belfast, Northern Ireland, United Kingdom, 2 Patrick G. Johnston Centre for Cancer Research, Queen's University Belfast, Belfast, United Kingdom, 3 Department of Pathology, Belfast Health and Social Care Trust, Grosvenor Road, Belfast, Northern Ireland, United Kingdom, 4 Department of Obstetrics and Gynaecology, Antrim Area Hospital, Northern Health and Social Care Trust, Antrim, Northern Ireland, United Kingdom

☉ These authors contributed equally to this work.
* u.mcmenamin@qub.ac.uk

**Data Availability Statement:** All relevant data are within the manuscript and its Supporting Information files.

## Abstract

### Background

To inform treatment decisions in women diagnosed with endometrial hyperplasia, quantification of the potential for concurrent endometrial cancer and the future risk of progression to cancer is required.

### Methods

We identified studies up to September 2018 that reported on the prevalence of concurrent cancer (within three months of endometrial hyperplasia diagnosis), or the incidence of cancer, identified at least three months after hyperplasia diagnosis. Random-effects meta-analyses produced pooled estimates and 95% confidence intervals (CIs).

### Results

A total of 36 articles were identified; 15 investigating concurrent and 21 progression to cancer. In pooled analysis of 11 studies of atypical hyperplasia, the pooled prevalence of concurrent endometrial cancer was 32.6% (95% CI: 24.1%, 42.4%) while no studies evaluated concurrent cancer in non-atypical hyperplasia. The risk of progression to cancer was high in atypical hyperplasia (n = 5 studies, annual incidence rate = 8.2%, 95% CI 3.9%, 17.3%) and only one study reported on non-atypical hyperplasia (annual incidence rate = 2.6%, 95% CI: 0.6%, 10.6%).

### Conclusions

Overall, a third of women with atypical hyperplasia had concurrent endometrial cancer, although the number of studies, especially population-based, is small. Progression to

**Funding:** OB Sanni was funded by a Queen's University Belfast International Studentship. https://www.qub.ac.uk/ ÚC McMenamin is funded by a Queen's University Belfast Vice Chancellor's Fellowship. https://www.qub.ac.uk/ HG Coleman is funded by a Cancer Research UK Career Establishment Award. https://www.cancerresearchuk.org/ The funders had no role in study design, data collection and analysis, decision to publish, or preparation of the manuscript.

**Competing interests:** The authors have declared that no competing interests exist.

cancer in atypical hyperplasia was high, but few studies were identified. Population-based estimates are required, in both atypical and non-atypical hyperplasia patients to better inform treatment strategies.

## Introduction

Endometrial cancer is the 6th most commonly diagnosed cancer in women worldwide, with the highest rates observed in developed countries, including the United States and Europe [1]. The incidence of endometrial cancer has increased in many countries over the past few decades [2], a trend which is hypothesized to be due to the rising prevalence of obesity, as well as shifts in female reproductive patterns [3]. Although there are currently no well-established screening programs for endometrial cancer, endometrial hyperplasia is a recognized precursor lesion of the most common type of endometrial cancer (endometrioid) and its detection offers opportunities for prevention [4].

Endometrial hyperplasia is diagnosed histologically in the presence of a proliferation of the endometrial glands resulting in an increase in gland-to-stroma ratio [5]. While endometrial hyperplasia can progress to endometrial cancer, the rate of progression depends on factors such as the degree of architectural abnormality and the presence or absence of nuclear atypia [6]. It is well-established that progression to endometrial cancer is higher in women with atypical compared with non-atypical hyperplasia but the magnitude of this risk is uncertain [7, 8], as most published studies have been conducted in single-center and tertiary referral centers which could overestimate risk in comparison to population-based studies. Endometrial hyperplasias (and even low-grade endometrioid cancers) can be conservatively managed by hormone therapies (e.g. oral and/ or intrauterine progestogens), especially among women who wish to maintain fertility. Currently if fertility preservation is not an issue, hysterectomy is generally recommended for women with atypical hyperplasia, due to the presumed significant risk of concurrent future endometrial cancer, and for women with persistent non-atypical hyperplasia [9, 10]. Accurately quantifying endometrial cancer risk in women diagnosed with endometrial hyperplasia is therefore crucial to inform shared decision making regarding the most appropriate clinical management strategies.

Recent clinical guidelines recommend that in women with endometrial hyperplasia who undergo conservative medical (non-surgical) management, endometrial biopsy should be undertaken at least every three months, until two consecutive negative biopsies are obtained, especially in patients with atypical hyperplasia[10]. A diagnosis of endometrial cancer within the first three months of an incident hyperplasia diagnosis is likely to reflect a concurrent finding that was missed at the initial investigation due to undersampling or was underdiagnosed by the reporting pathologist. Previous studies evaluating the prevalence of concurrent endometrial cancer in women with endometrial hyperplasia have reported wide-ranging estimates of between 30–50% [11–13]. However, prior systematic reviews have been restricted to women with atypical hyperplasia [11] and the results cannot be extrapolated to women diagnosed with the more common non-atypical endometrial hyperplasia (i.e. simple or complex hyperplasia using the 2003 World Health Organization (WHO) Classification [14]). The potential for concurrent endometrial cancer in women diagnosed with endometrial hyperplasia raises questions around the need for specialist pathology review, further endometrial sampling and/or hysterectomy in these patients, in addition to the identification of biomarkers that could assist in identifying those cases of endometrial hyperplasia which have the highest risk of concurrent cancer.

The aim of this study was to systematically review all studies which assessed the concurrent and future risk of endometrial cancer in women diagnosed with endometrial hyperplasia.

## Materials and methods

This systematic review was undertaken and reported in adherence to the Preferred Reporting Items for Systematic Reviews and Meta-Analyses (PRISMA) and the Meta-analysis of Observational Studies in Epidemiology (MOOSE) guidelines [15]. Three electronic databases including MEDLINE, EMBASE, and Web of Science were searched from 1994 up to September 2018 for relevant studies using a search-construct adapted for each database, see S1 Appendix. The databases were searched from 1994 as the WHO Classification of endometrial hyperplasia was established in that year based on criteria suggested by Kurman *et al.* [6].

Review articles, animal studies and articles written in languages other than English were excluded.

### Inclusion criteria

Titles and abstracts of identified articles were independently screened by at least two of four researchers (OBS, MTD, HGC, ÚMcM). At least two of the four reviewers then independently screened full texts to identify studies that met the pre-set inclusion criteria:

i. **Participants**: Women aged 18 and above who have received a diagnosis of endometrial hyperplasia

ii. **Interventions**: Report on risk of concurrent or future diagnosis of endometrial cancer, including the timeframes for follow-up. Endometrial cancer diagnosed within three months of a diagnosis of endometrial hyperplasia was categorized as concurrent disease while cancer diagnosed more than three months following endometrial hyperplasia diagnosis was regarded as disease progression. The three month cut-off timeline was chosen to account for diagnostic work-up [10].

iii. **Comparators**: Women aged 18 and above without a concurrent diagnosis of cancer or who did not progress to endometrial cancer

iv. **Outcome**: Risk of endometrial cancer

For a study to be eligible for inclusion, it had to include a minimum of 10 cases of endometrial hyperplasia to help ensure the inclusion of meaningful estimates of concurrent cancer risk or progression to cancer. Bibliographies of included studies were also reviewed. Any discrepancies throughout were discussed and resolved by agreement.

### Data extraction

Relevant information about study design, number of cases, controls, age and menopausal status of the study population, initial investigation method, cancer diagnosis method, WHO endometrial hyperplasia classification, intervention/ treatments received, time between diagnosis and treatment, and follow-up time were extracted in duplicate from full text articles. The Newcastle Ottawa Scale [16] was used to derive a quality score for each of the studies included in the review.

### Statistical analysis

Statistical analyses were conducted using STATA version 14. The proportion of prevalent endometrial cancer diagnosed in patients with endometrial hyperplasia was converted to a log

odds along with corresponding 95% confidence intervals (CIs), the random effects model was applied and the pooled log odds and 95% CIs back-transformed [17]. Incidence rates of endometrial cancer were calculated for studies that examined endometrial hyperplasia progression to endometrial cancer based upon the number of cases and the person years within each study, from multiplying the number of endometrial hyperplasia cases by the mean (two studies) or median (eight studies) follow-up time. Similarly, incidence rates were converted to the log incidence rates and corresponding 95% CIs and random effects meta-analysis was applied. Heterogeneity of studies included in meta-analyses were assessed using the $I^2$ statistic [18]; $I^2$ values of 25%, 50% and 75% are typically interpreted as low, moderate and high heterogeneity, respectively. Sub-group analyses were planned by type of hyperplasia and where possible, by menopausal status and method of hormone therapy administration. Separate sensitivity analyses were performed restricting to studies with a quality score of less than five and more than or equal to five and by systematically removing each study in turn in order to determine its effect on the overall pooled estimates. We conducted additional sensitivity analyses investigating the impact of duration of follow-up in studies of endometrial hyperplasia progression to endometrial cancer (less than 24 months average follow-up, more than or equal to 24 months average follow-up). It was impossible to calculate rates per 1,000 person-years in 11 studies of future risk of endometrial cancer due to unavailability of information regarding follow-up and the results from these studies were narratively summarized. Publication bias was evaluated using a funnel plot, showing the standard error of log incidence against each study's prevalence or incidence rate.

## Results

After application of our search strategy, and removal of duplicates, a total of 1,587 titles and abstracts were reviewed to determine potentially relevant studies for inclusion. After title and abstract review, 148 full text and abstracts were reviewed, of which 70 were excluded (see Fig 1). A further 44 articles that assessed endometrial cancer risk were excluded as it was unclear whether cancer was identified within three months of endometrial hyperplasia diagnosis. Results from these 44 studies are outlined in S1 Table and S2 Table; S1 Table details 21 studies in which it was unclear if endometrial cancer was diagnosed within three months of hyperplasia diagnosis while S2 Table lists 23 studies that investigated concurrent endometrial cancer according to the author's own definitions, but did not report the time between endometrial biopsy and hysterectomy. Four further articles were identified for inclusion from review of reference lists. Two articles from the supplementary material did not meet the inclusion criteria for concurrent endometrial cancer; however, they were included in the review as they met the inclusion criteria for incident endometrial cancer. This left a total of 36 articles included, see Fig 1.

Characteristics of the included studies are outlined in Tables 1–3. Fifteen of the included articles assessed concurrent endometrial cancer (Table 1) and 21 assessed the risk of future endometrial cancer in women with endometrial hyperplasia (Tables 2 and 3). Fifteen studies were conducted in Europe, 12 in Asia, eight in North America (one study included patients from the US and Europe) and one in Australia. The majority of studies were single or two center studies, and only one study was population-based. Endometrial biopsy or dilatation and curettage was the most common method of endometrial hyperplasia diagnosis; however, some studies used other methods such as hysteroscopic resection [19]. Most studies were classified as being of low-to-moderate quality, with quality scores ranging from four to six out of a total of nine.

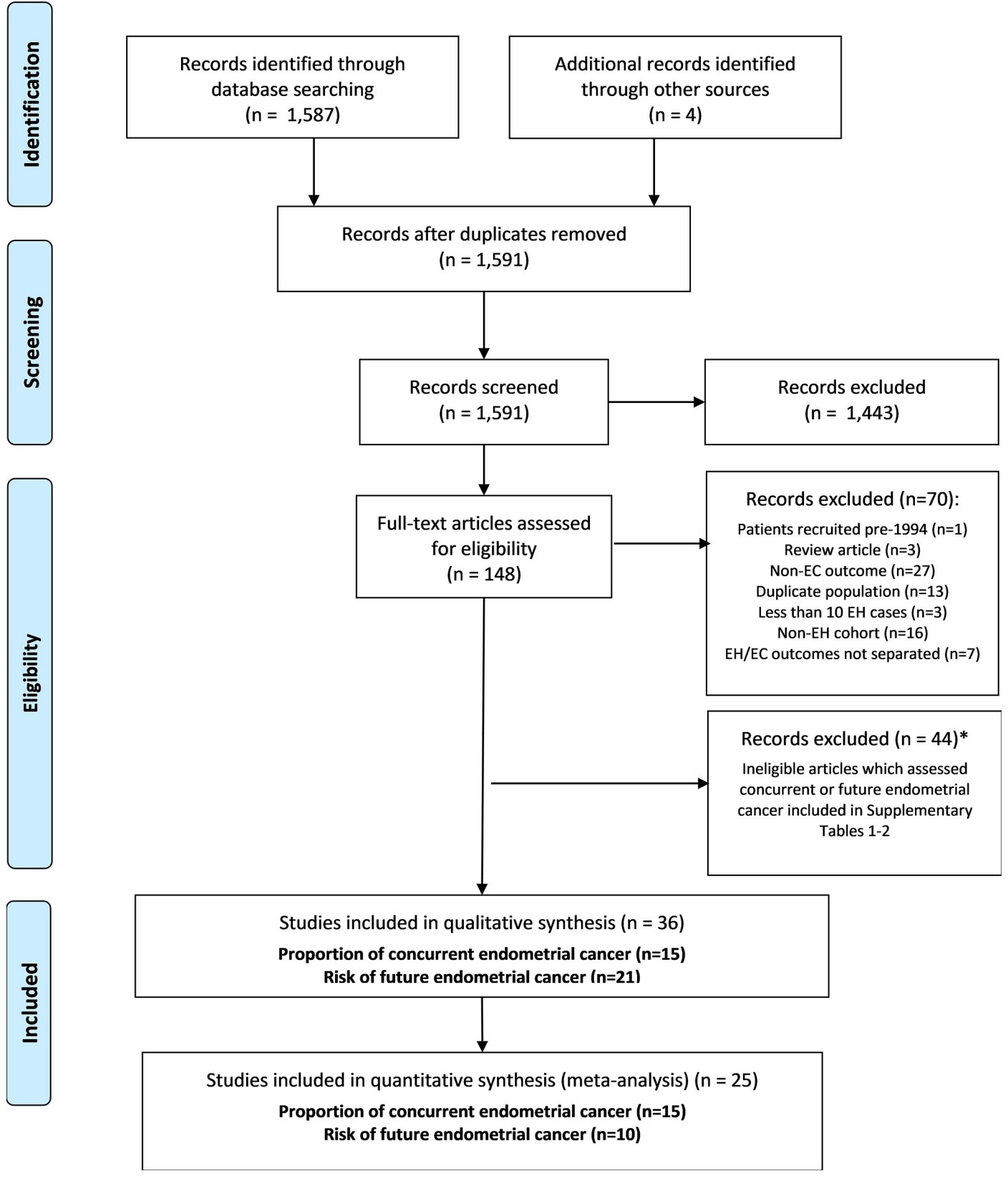

**Fig 1. Flow diagram of study selection for the systematic review of prevalence of concurrent and risk of future endometrial cancer among patients with endometrial hyperplasia.** *Two articles outlined in the supplementary material did not meet the criteria for inclusion in the concurrent endometrial cancer investigation but were included in the review as they met the inclusion criteria for the incident endometrial cancer investigation. EC = endometrial cancer, EH = endometrial hyperplasia.

## Proportion of concurrent endometrial cancer diagnosed in women with endometrial hyperplasia

The prevalence of endometrial cancer diagnosed in endometrial hyperplasia patients was assessed in 15 studies [19–33] including 1,496 women (range 17–289), Table 1. The majority of studies investigated atypical hyperplasia only (n = 11) [19, 20, 22, 23, 25–29, 31, 33] and four studies investigated any type of hyperplasia [21, 24, 30, 32]. Hysterectomy was the method of follow-up investigation in all studies. None of the identified studies evaluated concurrent endometrial cancer in women with non-atypical hyperplasia only. The prevalence of concurrent endometrial cancer diagnosed within three months of an endometrial hyperplasia diagnosis ranged from 5.9% to 53.1%.

The overall pooled proportion of concurrent endometrial cancer in all 15 studies was 32.1% (95% CI: 26.1%, 40.0%) and high heterogeneity ($I^2$ = 87.6%) was observed (Fig 2). In a pooled analysis of 11 studies, the pooled proportion of concurrent endometrial cancer in women with atypical hyperplasia was 32.6% (95% CI 24.1%, 42.4%) and high heterogeneity was observed ($I^2$ = 87.9%), as shown in Fig 2. A slightly lower proportion concurrent endometrial cancer was observed after pooling the four studies that included a mixture of atypical and non-atypical hyperplasia (30.8%, 95% CI: 18.6%, 46.4%) but heterogeneity was high ($I^2$ = 88.3%). The funnel plot for studies evaluating concurrent endometrial cancer did not appear to show any strong evidence of a lack of symmetry and therefore was not indicative of publication bias (S1 Fig).

## Sensitivity and subgroup analysis

Results remained similar, and heterogeneity remained high, in sub-group analysis by study quality and in sensitivity analyses excluding individual studies, see S3 Table. All studies included both pre and postmenopausal women (three studies did not specify [22, 26, 29]) but there were too few stratified results by menopausal status to allow the conduct of sub-group analysis.

## Risk of future endometrial cancer diagnosis in women with endometrial hyperplasia

A total of 21 studies including 2,495 women assessed the risk of future endometrial cancer in women with endometrial hyperplasia [34–54], see Tables 2 and 3. Nine of the studies [35–38, 45, 46, 50, 51, 54] investigated atypical hyperplasia only, three investigated non-atypical hyperplasia [39, 43, 44], while nine studies investigated any type of hyperplasia [34, 40–42, 47, 49, 48, 52, 53].

Incidence rates per 1000 person-years were calculated for 10 studies that assessed the risk of future endometrial cancer and reported mean or median follow-up time (Table 2). Two studies reported mean follow-up [38, 45], eight reported median follow-up [34, 39, 41, 46, 47, 50–52]. The rate of future endometrial cancer in women with endometrial hyperplasia ranged from 4.3 to 287.9 per 1000 person-years with follow-up time ranging from three months to 23 years.

Pooled analysis of five individual studies (including six estimates) including 169 women with atypical hyperplasia showed that the incidence rate of endometrial cancer was 82.3 per

**Table 1. Characteristics of studies which assessed the prevalence of concurrent endometrial cancer in women with endometrial hyperplasia (n = 15).**

| Author, Year Location | Study population | Study design | Recruitment period | No. EH cases | No EC cases | % concurrent EC | Age (mean) (SD) | Method of initial investigation | Time from biopsy to hysterectomy | EH investigated | Quality score |
|---|---|---|---|---|---|---|---|---|---|---|---|
| Agostini, 2003 France [19] | Hôspital la Conception | Single-center retrospective cohort | 1994–2001 | 17 | 1 | 5.9 | 54.4 | Hysteroscopic resection | 3 weeks | AH | 4 |
| Bilgin, 2004 Turkey [20] | Uludag University Hospital | Single-center retrospective cohort | Not reported | 46 | 11 (1 SH, 10 CAH) | 23.9 | 48.9 (8.3) | D&C, pipelle biopsy | Within 6 weeks | AH | 4 |
| Dolanbay, 2015 Turkey [21] | Erciyes University | Single-center retrospective cohort | 2009–2013 | 82 (40 CAH, 13 SAH, 20 SH, 9 CH) | 39 (28 CAH, 7 SAH, 4 CH) | 47.5 | 54 (8.7) | Pipelle biopsy, D&C | < 6 weeks | All | 4 |
| Gungorduk, 2014 Turkey [22] | Tepecik Hospital, Bulent Ecevit University, Sisli Etfal Hospital, Istanbul Education & Research Hospital | Multicenter retrospective cohort | 1996–2003 | 128 | 68 | 53.1 | 54.2 (30–82) | Pipelle biopsy, D&C | 2 weeks (median, range 1–4) | CAH | 4 |
| Hahn, 2010 Korea [23] | Cheil General Hospital and Women's Healthcare Center | Two-center retrospective cohort | 1999–2008 | 126 (24 SAH, 102 CAH) | 13 | 10.3 | 45.4 (6.6) | D&C, biopsy, hysteroscopic polypectomy | < 12 weeks | AH | 4 |
| Karamursel, 2005 Turkey [24] | Ankara Maternity & Women's Health Teaching Hospital, Hacettepe University Hospital | Two-center retrospective cohort | 1990–2003 | 204 (56 AH, 148 NAH) | 43 (35 AH, 8 NAH) | 21.1 | 57.4 (range 28–87) | D&C | Within 1 month | All | 5 |
| Kimura, 2003 Japan [25] | Osaka Medical Center for Cancer and Cardiovasucular Diseases. | Single-center retrospective cohort | 1992–2002 | 33 | 9 | 27.3 | 51.7 (EC cases) 50.0 (EC non-cases) | Biopsy | 8 weeks | AH | 4 |
| Lai, 2014 Taiwan [26] | National Defense Medical Center, Taipei | Single-center retrospective cohort | Not reported | 61 | 14 | 23 | Not reported | Biopsy | Within 3 months | AH | 4 |
| Merisio, 2005 Italy [27] | University of Parma, Policlinico san Matteo Hospital | Two–center retrospective cohort | 1992–2003 | 70 | 30 | 42.9 | 55.5 (11.9) (range 38–80) | Pipelle biopsy, D&C | 2–8 weeks | AH | 4 |
| Morotti, 2012 Italy [28] | San Martino Hospital | Single-center retrospective cohort | 2000–2011 | 66 | 35 | 53 | 58.5 (median, range 34–76) | Biopsy, D&C, pelvic lymph node dissection | < 6 weeks | AH | 4 |
| Mutter, 2008 USA [29] | Gynecologic Oncology Group trial 167A | Multicenter retrospective cohort | 1998–2003 | 289 | 124 | 42.9 | Not reported | Pipelle biopsy, D&C | Within 3 months | AH | 5 |

*(Continued)*

**Table 1.** (Continued)

| Author, Year Location | Study population | Study design | Recruitment period | No. EH cases | No EC cases | % concurrent EC | Age (mean) (SD) | Method of initial investigation | Time from biopsy to hysterectomy | EH investigated | Quality score |
|---|---|---|---|---|---|---|---|---|---|---|---|
| Pavlakis, 2010 Greece [30] | IASO Women's Hospital | Single-center retrospective cohort | Not reported | 83 (4 SH, 19 CH, 27 CAH) | 33 (31 CAH, 2 CH) | 39.7 | 35–67 (range) | D&C | Within 12 weeks | SH, CH, CAH | 4 |
| Rakha, 2012 UK [31] | Nottingham University Hospital | Single-center retrospective cohort | 1987–2011 | 219 | 59 | 27 | Not reported | Biopsy | Within 3 months | AH | 4 |
| Salman, 2010 Turkey [32] | Hacettepe University Hospital | Single-center retrospective cohort | 2007–2009 | 49 (3 SAH, 12 CH, 34 CAH) | 9 (CAH) | 18.4 | 51.5 (range 36–79) | Biopsy | Within 2 weeks | All | 4 |
| Valenzuela, 2003 Spain [33] | University Hospital Principe de Asturias | Single-center retrospective cohort | 1988–2001 | 23 | 12 | 52.2 | 52 (range 30–83) | Biopsy, fractional curettage | 10.5 weeks | AH | 4 |

EH- endometrial hyperplasia, EC- endometrial cancer, AH- atypical hyperplasia, SH-simple hyperplasia, CAH- complex atypical hyperplasia, D&C- dilation and curettage, SAH- simple atypical hyperplasia, CH-complex hyperplasia, NAH- non-atypical hyperplasia

1,000 person-years (95% CI 39.3, 172.6), equivalent to 8.2% per year, with high heterogeneity observed ($I^2$ = 70%), Fig 3. Only one study included only non-atypical hyperplasia patients [39] and reported a much lower rate of progression to cancer of 26.3 per 1,000 person-years (95% CI: 6.6, 105.6), or 2.6% per year. Among 4 studies (including 5 estimates), that included both atypical and non-atypical hyperplasia patients, the pooled rate of progression to cancer was 12.4 per 1,000 person-years (95% CI: 6.2, 24.9), or 1.2% per year (again incorporating high heterogeneity, $I^2$ = 81%), Fig 3. After pooling results from all 10 studies (including 12 estimates), with 1,400 women with any type of endometrial hyperplasia, the incidence rate for progression to endometrial cancer was 31 per 1,000 person-years (95% CI: 14.7, 65.6), equivalent to 3.1% per year. There was high heterogeneity ($I^2$ = 90%), Fig 3.

Although it was impossible to calculate rates per 1,000 person-years due to unavailability of information regarding follow-up time in 11 studies including 1,095 women [35–37, 40, 42, 43, 44, 48, 49, 53, 54], the total percentage of patients who progressed from hyperplasia to cancer was calculated and ranged from 1.3% to 31.6%, see Table 3. Follow-up time in these studies ranged from three months to 20 years. We identified only one population-based study that assessed future endometrial cancer risk at least three months after hyperplasia diagnosis; Lacey et al. [53] reported that the relative risk was three times higher in endometrial hyperplasia patients compared to patients with disordered proliferative endometrium after one year, and that the risk was more marked for atypical hyperplasia [53]. However, due to the nested case-control study design used, we were unable to include this study in our meta-analysis, as we could not calculate an incident rate ratio. There was no strong evidence of a lack of symmetry based on the funnel plot of studies investigating progression to endometrial cancer, which therefore was not indicative of publication bias (S2 Fig).

## Sensitivity and subgroup analysis

In analysis by study quality, the risk of progressing to cancer was higher when restricting to studies deemed to be of lower quality (6.6% per year, 95% CI 1.23%, 33.86%) but heterogeneity

**Table 2. Characteristics of studies which assessed the risk of future endometrial cancer in women with endometrial hyperplasia and for which rates per person-years were calculated (n = 10).**

| Author, Year Location | Study population | Study design | Recruitment period | No. EH cases | No EC cases | % progression | Rates per 1000 person-years | Age (mean) (SD) | Method of initial investigation | Method of follow-up investigation | Follow-up time | EH investigated | Quality score |
|---|---|---|---|---|---|---|---|---|---|---|---|---|---|
| Baak 2005 Europe/USA [34] | 6 European centers, 2 US centers | Multicenter prospective cohort | Not reported | 477 (289 SH, 65 CH, 67 SAH, 56 CAH) | 264 (2 SH, 6 CH, 5 SAH, 11 CAH) | 5 | 12.6 | Not reported | Biopsy, D&C | Biopsy | 48 months (median) (range 13–120) | All | 6 |
| Brownfoot, 2014 Australia [35] | Royal Women's Hospital, Melbourne | Single-center retrospective cohort | 1999–2012 | 42[a] 19[b] | 2[a] 6[b] | 4.8[a] 31.6[b] | 23.8[a] 105.3[b] | Premenopausal: 37 years (7.6) Postmenopausal: 61 years (11.4) | Biopsy, hysteroscopy | Hysteroscopy, biopsy, hysterectomy | 24 months (median)[a],36 months (median)[b] | CAH | 5 |
| Edris, 2007 Canada [36] | University of Western Ontario | Single-center prospective cohort | 1990–2005 | 16 | 1 | 6.3 | 12.5 | 24–78 (range) | Hysteroscopy | Hysteroscopy, hysterectomy | 5 years (median, range 1.5–12) | AH | 5 |
| Gallos, 2013 UK [37] | Birmingham Women's Hospital | Single-center comparative study | 1998–2007 | 250[c] 94[d] | 6[c] 4[d] | 2.4[c] 4.3[d] | 4.3[c] 5.9[d] | 52.7 (10.6)c 48.5 (11.6)[d] | Biopsy | Hysterectomy | 66.9 months (median, range 12–148.2)[c], 87.2 months (median, range 13.2–162)[d] | CH, CAH | 5 |
| Garuti, 2005 Italy [38] | Lodi Hospital | Single-center prospective cohort | 1997–2003 | 24 (20 SH, 4 CH)[e] | 2 (1 SH, 1 CH) | 8.3 | 26.3 | 62.3 (8.7)[f] | Hysteroscopy, biopsy | Hysterectomy | 38 months (median, range 12–60) | SH, CH | 5 |
| Gonthier, 2015 France [39] | French gynecological units | Multicenter retrospective cohort | 2001–2013 | 32 | 9 | 13.2 | 70.3 | 34 (33.6) (range 23–42) | Biopsy D&C | Biopsy D&C | 48 months (mean, range 7–121) | AH | 4 |
| Horn, 2004 Germany [40] | University of Leipzig | Single-center retrospective cohort | Not reported | 215 (208 CH, 7 AH) | 5 (2 CH, 3 AH) | 2.3 | 58.1 | Not reported | Fractional curettage | Fractional curettage | 4.8 months (median, range 3–22 months) | CH, AH | 5 |
| Mentrikoski, 2012 USA [41] | University of Virginia, and Emory University | Two-center retrospective cohort | Not reported | 18 | 3 | 16.7 | 133.3 | 38 (range, 25–39)[a] 57 (range, 50–74)[b] | Biopsy, D&C | Hysterectomy | 15 months (mean) | CAH | 5 |

*(Continued)*

Table 2. (Continued)

| Author, Year Location | Study population | Study design | Recruitment period | No. EH cases | No EC cases | % progression | Rates per 1000 person-years | Age (mean) (SD) | Method of initial investigation | Method of follow-up investigation | Follow-up time | EH investigated | Quality score |
|---|---|---|---|---|---|---|---|---|---|---|---|---|---|
| Steinbakk, 2011 Norway [42] | Stavanger University Hospital | Single-center retrospective cohort | 1980–2004 | 152 (114 SH, 4 SAH, 26 CH, 8 CAH) | 11 (4 SH, 1 SAH, 2 CH, 4 CAH) | 7.2 | 15.2 | 53 years (range 21–88) | D&C | Not reported | 57 months (median, range 12–283) | All | 4 |
| Tierney, 2014 USA [43] | Los Angeles County + USC Medical Center | Single-center retrospective cohort | 2003–2011 | 61 | 6 | 9.8 | 287.9 | 72% were <40 years old. | Biopsy | Biopsy | 4.1 months (median, range 1.1, 29.2) | CAH | 4 |

EH- endometrial hyperplasia, EC- endometrial cancer, AH- atypical hyperplasia, SH- simple hyperplasia, CAH- complex atypical hyperplasia, D&C- dilation and curettage, SAH- simple atypical hyperplasia, CH-complex hyperplasia, NAH- non-atypical hyperplasia

[a] Premenopausal women

[b] Postmenopausal women

[c] LNG-IUS (levonorgestrel intrauterine system) treated group

[d] Oral progesterone-treated group

[e] Cohort comprises patients with a prior diagnosis of breast cancer

[f] Includes two patients who underwent hysterectomy after EH diagnosis

[g] Patients received first post-treatment biopsy after at least 3 months of progestin therapy.

**Table 3. Characteristics of studies which assessed the risk of future endometrial cancer in women with endometrial hyperplasia which did not report mean or median follow-up and for which rates per person-years calculations were not possible (n = 11).**

| Author, Year Location | Study population | Study design | Recruitment period | No. EH cases | No EC cases | % progression | Age (mean) | Method of initial investigation | Method of follow-up investigation | Follow-up time | EH investigated | Quality score |
|---|---|---|---|---|---|---|---|---|---|---|---|---|
| Anastasiadis, 2000 Greece [44] | General Hospital of Alexandroupolis | Single-center retrospective cohort | 1986–1998 | 294 (258 NAH, 36 AH) | 4 (1 SH, 2 CH, 1 AH) | 1.4 | Not reported | D&C, hysteroscopy | D&C | At least 6 months | All | 4 |
| Chen, 2016 China [45] | Peking Union Medical College Hospital | Single-center retrospective cohort | 2000–2011 | 16 | 2 | 12.5 | 20–42 (range) | Biopsy, D&C, hysteroscopy | Hysterectomy | At least 3 months | CH | 5 |
| Hecht, 2005 Israel [46] | Beth Israel Hospital | Single-center retrospective cohort | 1998–2000 | 84 (21 CAH, 15 CH, 48 SH) | 8 (5 CAH, 2 CH, 1 SH) | 9.5 | Not reported | Biopsy, D&C | Hysterectomy, endometrial sampling or clinical follow-up | Not reported | All | 4 |
| Inversen, 2018 Denmark [47] | Regional Hospital Holstebro, Aarhus University Hospital | Two-center retrospective cohort | 2000–2005 | 114 | 17 | 14.9 | 59.1 | Biopsy, D&C, Trans cervical hysteroscopic endometrial resection | Hysterectomy | 9–14 years | CH | 5 |
| Lacey, 2008 USA [48] | Kaiser Permanente Northwest | Population-based nested case-control | 1970–2003 | 368 (127 cases and 241 controls) | 127 | Not applicable | 52 | Biopsy, D&C | Hysterectomy | 6.7 years (median, range 1–24.5) | All | 7 |
| Minig, 2011 Italy [49] | European Institute of Oncology, Milan | Single-center prospective cohort | 1996–2009 | 20 | 1 | 5 | Not reported | Pipelle biopsy, D&C, hysteroscopy | Hysterectomy | At least 6 months | AH | 5 |
| Orbo, 2000 Norway [50] | Northen Norway | Multicenter retrospective cohort | Not reported | 68 (9 SH, 10 CH, 13 SAH, 36 CAH) | 18 (1 CH, 1 SAH, 16 CAH) | 26 | 28–77 (range) | D&C | Hysterectomy | 10–20 years[a] | All | 5 |
| Simpson, 2014 Canada [51] | Princess Margaret Hospital and Odette Cancer Center | Two-center retrospective cohort | 2000–2011 | 19 | 6 | 31.6 | All <45 | Biopsy, D&C | Hysterectomy | 39 months (median, range 5–128)[b] | CAH | 5 |
| Tabata, 2001 Japan [52] | Yamada Red Cross Hospital | Single-center prospective cohort | 1989–1996 | 77 (48 SH, 17 CH, 1 SAH, 11 CAH) | 1 (CAH) | 1.3 | 47.2 (SH), 46.6 (CH), 52.0 (SAH), 47.9 (CAH) | D&C | D&C | At least 1 year | All | 4 |
| Ushijima, 2007 Japan [53] | Japan Gynecologic Cancer Study Group | Multicenter prospective trial | Not reported | 17 | 2 | 11.8 | 20–39 (range) | D&C | Hysterectomy | At least 8 weeks | AH | 5 |

(Continued)

**Table 3.** (Continued)

| Author, Year Location | Study population | Study design | Recruitment period | No. EH cases | No EC cases | % progression | Age (mean) | Method of initial investigation | Method of follow-up investigation | Follow-up time | EH investigated | Quality score |
|---|---|---|---|---|---|---|---|---|---|---|---|---|
| Wheeler, 2007 USA [54] | Johns Hopkins Hospital | Single-center retrospective cohort | Not reported | 18 | 2 | 11.1 | 34 (median, range 24–47)[c] 61 (median, range 50–77)[d] | Biopsy, D&C | Biopsy, hysterectomy | 3–25 months | CAH | 4 |

EH- endometrial hyperplasia, EC- endometrial cancer, AH - atypical hyperplasia, SH-simple hyperplasia, CAH- complex atypical hyperplasia, D&C- dilation and curettage, SAH- simple atypical hyperplasia, CH-complex hyperplasia, TVUS- transvaginal ultrasound scan, NAH- non-atypical hyperplasia

[a] Follow-up is among EH patients who developed endometrial cancer

[b] Follow-up time also includes grade 1 EC patients (n = 25)

[c] Premenopausal women

[d] Postmenopausal women

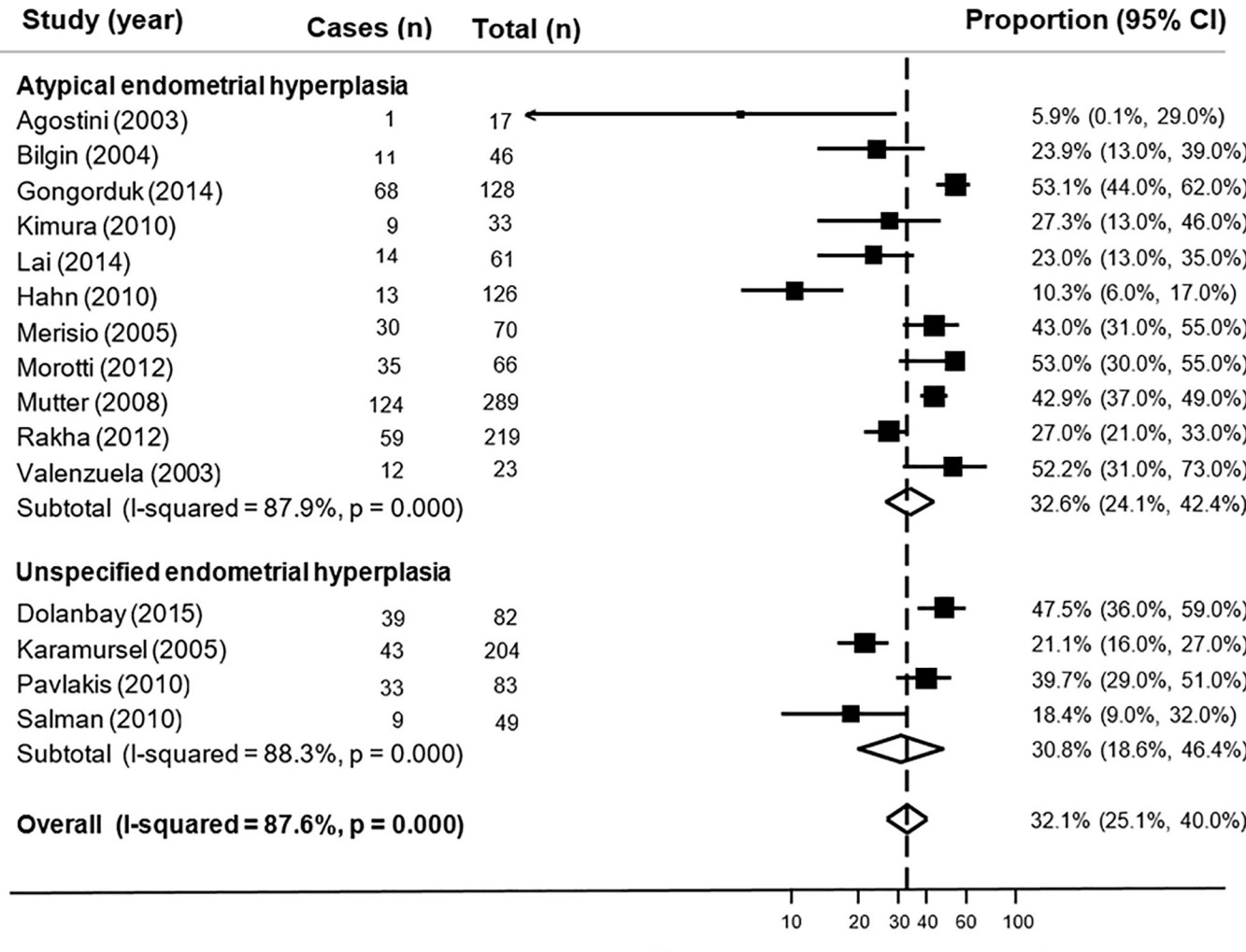

**Fig 2. Forest plot of proportion of concurrent endometrial cancer diagnosed within three months of endometrial hyperplasia diagnosis.**

remained high, see S3 Table. Results were largely similar (and heterogeneity remained high) in sensitivity analyses excluding individual studies (S3 Table). Most studies included both pre and post-menopausal women but one study included only postmenopausal women [39]; in another, we could only include results for premenopausal women [39]. Brownfoot et al. [50] presented results for atypical hyperplasia patients stratified by menopausal status and showed that postmenopausal women were more likely to progress to cancer compared to premeno-pausal women (10.5% versus 2.4% progression per year, respectively) but too few other studies stratified results by menopausal status and therefore a sub-group analysis was precluded. Six studies reported information on exposure to progestogen therapy [38, 45, 47, 50, 51, 52], with women treated either orally or via an intrauterine device. Gallos et al. [47] provided results by progestogen therapy type and found that women with complex non-atypical or atypical hyper-plasia who were treated with oral progestogens were more likely to progress to cancer com-pared to women treated with the levonorgestrel-releasing intrauterine system (5.9% versus 4.3% progression per year, respectively).

Results from sensitivity analyses stratifying on average duration of follow-up are presented in S3 Fig and S4 Fig. Pooled analysis of three studies with an average follow-up of less than 24

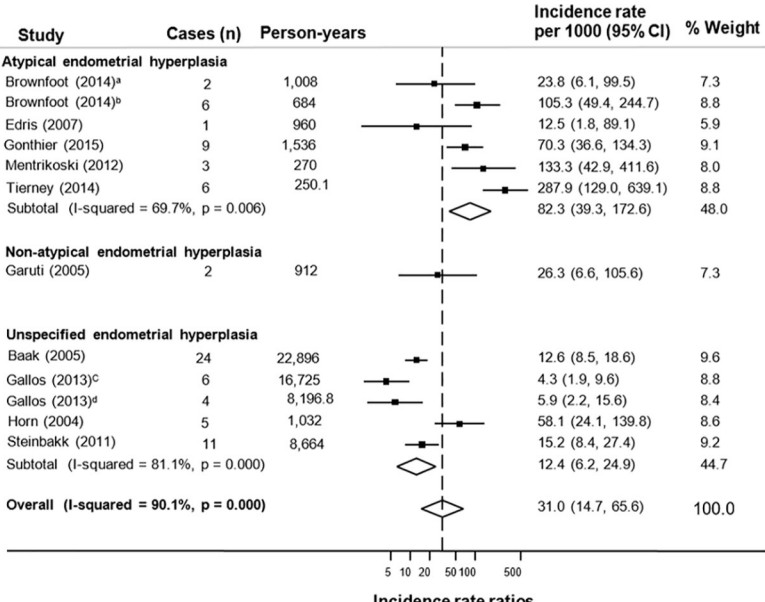

**Fig 3. Forest plot of incidence rates of endometrial cancer diagnosed after three months of endometrial hyperplasia diagnosis.** [a]Premenopausal women, [b]Postmenopausal women, [c]LNG-IUS (levonorgestrel intrauterine system) treated group, [d]Oral progesterone-treated group.

months, including 121 women with atypical hyperplasia, showed that the incidence rate of endometrial cancer was 107 per 1,000 person-years (95% CI 27.6, 414.9), equivalent to 10.7% per year, with high heterogeneity observed ($I^2$ = 78.3%), S3 Fig. Among three studies with an average follow-up of more than or equal to 24 months, pooled analysis including 67 women with atypical hyperplasia showed a lower incidence rate of endometrial cancer (66.1 per 1,000 person-years, 95% CI 30.4, 144), equivalent to 6.6% per year, with moderate heterogeneity observed ($I^2$ = 49.5%), S4 Fig. Only one study evaluated progression to endometrial cancer specifically in women with non-atypical endometrial hyperplasia [39], which precluded meta-analysis. Four studies that included both atypical and non-atypical hyperplasia patients, had an average follow-up of more than or equal to 24 months (n = 973 women) and pooled analysis showed that the incidence rate of endometrial cancer was 9.2 per 1,000 person-years (95% CI 5.4, 15.9), equivalent to 0.9% per year, with high heterogeneity observed ($I^2$ = 87.7%), S4 Fig. We identified no studies that investigated both atypical and non-atypical hyperplasia patients with an average follow-up of less than 24 months.

## Discussion

Our meta-analysis of 15 studies showed that approximately 32% of women with endometrial hyperplasia received a concurrent diagnosis of endometrial cancer, with the majority of studies only including women diagnosed with atypical hyperplasia (n = 11). The overall future risk of progression to endometrial cancer was 3% per year, and this was markedly higher for patients with atypical hyperplasia (8%). However, high heterogeneity was observed in pooled analyses and the majority of the studies were of relatively low quality and few specifically included pre-menopausal women.

Our findings regarding concurrent endometrial cancer risk are similar to those reported in an earlier systematic review which investigated the risk of concurrent endometrial cancer in hysterectomy specimens comparing three methods of endometrial sampling (curettage,

hysteroscopically guided biopsy and hysteroscopic resection) [12]. In that review, the rates of concurrent endometrial cancer were 32.7%, 45.3% and 5.8% following an endometrial hyperplasia diagnosis by curettage, hysteroscopically guided biopsy and hysteroscopic resection, respectively. However, the review was restricted to studies published up to 2013, included only atypical hyperplasia patients, and the authors did not specify a cut-off time for endometrial cancer identification. In another previous systematic review which included a sub-set of endometrial hyperplasia patients (atypical hyperplasia occurring within a polyp), a much lower proportion were found to have concurrent endometrial cancer (5.6%) [55]. However, unlike the current study, estimates from that review are likely not generalizable to women with endometrial hyperplasia not confined to polyps. More recently, a meta-analysis by Travaglino et al. [56] of studies conducted after 2008, found that women with atypical hyperplasia were over 11 times more likely to have coexistent cancer, which the authors defined as cancer occurring within a year of hyperplasia diagnosis, compared to women with non-atypical hyperplasia and the results were similar when evaluating studies using the alternative endometrial intraepithelial neoplasia (EIN) system for classification of the precursor lesions of endometrioid cancer. A further systematic review investigated occult cancer risk in complex non-atypical endometrial hyperplasia and found that cancer was present in 2% of surgical specimens in women diagnosed with simple endometrial hyperplasia and 12.4% in women diagnosed with complex endometrial hyperplasia, suggesting that complexity of glandular architecture is an important marker of occult cancer [57]. However, unlike our systematic review, there was no time cut-off used to define occult cancer and the time to hysterectomy ranged from 4 days to 7 years within the individual studies, meaning that the outcomes may have represented progression to cancer rather than coexistent cancer. Underdiagnosis of endometrial cancer can result in inadequate staging and potentially suboptimal treatment; therefore, high-quality, population-based studies are required to more accurately determine the proportion of concurrent cancer in women diagnosed with both atypical and non-atypical endometrial hyperplasia.

In pooled analysis of studies evaluating progression to endometrial cancer, we identified a high rate of progression in women diagnosed with atypical hyperplasia (8.2% per year), while the risk among women with non-atypical hyperplasia was much lower at 2.6% per year but only one study was identified [39]. The risk of progressing from atypical hyperplasia to endometrial cancer was higher in studies with less than 2 years average follow-up (10.7% per year), suggesting that closer surveillance may be necessary during this period, however risk of progression was still high among studies with follow-up of more than 2 years on average (6.6% per year). A previous systematic review of women diagnosed with atypical hyperplasia or endometrial cancer reported that 15% of those who used oral progestogens had endometrial cancer with at least myometrial invasion on the hysterectomy specimen after a mean follow-up of 49 months, however estimates for atypical hyperplasia were not reported [7]. In another review including 12 studies and restricted to atypical hyperplasia, 2.7% of patients who used oral progestogen for more than six months progressed to endometrial cancer [8]. These earlier systematic reviews reported rates of progression to endometrial cancer in terms of crude percentages rather than the more clinically meaningful percentage progression per year that we report. Moreover, most prior studies were conducted in single-center and tertiary referral centers, which could overestimate risk in comparison to population-based studies. One of the few prior population-based studies, conducted by Reed et al. [58], demonstrated a much lower risk of endometrial cancer (0.9% per year) among 1,443 complex or atypical endometrial hyperplasia patients identified from a health insurance Group Health plan, in Washington State, USA. This study was included in the supplemental material of our review as the minimum time between diagnosis of hyperplasia and cancer was less than three months. Future population-based studies of endometrial cancer progression in women with endometrial hyperplasia are

therefore required to provide robust risk estimates. Furthermore, considering the rising incidence of endometrial cancer in younger women [2], future studies should aim to specifically include premenopausal hyperplasia patients.

The reasons underlying the high proportion of concurrent endometrial cancers in women diagnosed with endometrial hyperplasia, especially atypical hyperplasia, are complicated but are likely to include undersampling and suboptimal histopathological diagnosis. Many endometrial biopsies, especially outpatient pipelle biopsies and curettages, are scanty with little tissue represented and the absence of invasion or stroma on slides makes the diagnosis more challenging and risks potentially missing the most significant lesion. Therefore, pathologists may be less likely to make a firm cancer diagnosis.

In addition, it is well known amongst pathologists that there is significant interobserver variability in the reporting of endometrial hyperplasias, including the distinction between benign lesions and non-atypical hyperplasias at the lower end of the spectrum, the distinction between non-atypical and atypical hyperplasia and at the upper end of the spectrum the distinction between atypical hyperplasia and low-grade endometrioid carcinoma [59–61]. The recently published International Society of Gynecological Pathologists endometrial cancer recommendations discuss that in some cases the distinction between atypical hyperplasia and low-grade endometrioid carcinoma is problematic and states "If the morphologic features are suspicious but do not fully meet the criteria for endometrioid carcinoma, this concern should be communicated descriptively in the pathology report rather than being classified as atypical hyperplasia without further comment. There are no diagnostically useful biomarkers to distinguish between atypical hyperplasia and low-grade endometrioid carcinoma" [62]. An additional confounding factor is that in many institutions, most endometrial biopsies are reported by non-specialist pathologists where there is likely to be more interobserver variability than amongst specialist gynecological pathologists.

To our knowledge, this is the first systematic review examining both the prevalence of concurrent and risk of future endometrial cancer in women with endometrial hyperplasia. The use of three large databases and robust screening of articles by at least two independent reviewers minimized the potential for selection bias. Other strengths include its large size and comprehensive inclusion of all endometrial hyperplasia types. We conducted novel meta-analyses for assessment of future endometrial cancer risk and reported the more clinically meaningful percentage progression per year in endometrial hyperplasia patients rather than total percentage of progression. We applied a strict cut-off of three months to distinguish between concurrent and future endometrial cancer on the basis of clinical recommendations for hyperplasia (in particular atypical hyperplasia) follow-up investigations [10].

This review was limited by the high degree of heterogeneity across the included studies, which is likely due to differences in study populations, methods used to diagnose endometrial hyperplasia, as well as the wide variation in the time between biopsy and hysterectomy, which ranged from three months to 12 years in studies of endometrial cancer progression. Few studies included results specifically for non-atypical endometrial hyperplasia, which limited subgroup analyses by hyperplasia type. We were unable to conduct sub-group analyses by menopausal status because the majority of included studies did not stratify by menopausal status. Moreover, few studies included sufficiently detailed information on patient management, including progestogen use, which limited conclusions around the impact of fertility-sparing therapies on cancer risk and the lack of information on these treatments may have affected prevalence and incidence estimates. Considering the rising incidence of endometrial cancer in both pre and postmenopausal women [63], temporal population-based estimates of progression to cancer in endometrial hyperplasia are warranted and such studies should account for treatments and other important clinical factors.

## Conclusion

In conclusion, a third of women with atypical endometrial hyperplasia were found to receive a concurrent diagnosis of endometrial cancer; however most studies were small in size and no study reported estimates specifically for non-atypical hyperplasia patients. The risk of progression from atypical hyperplasia to endometrial cancer was 8% per year, but few studies were identified. Population-based studies, which include both atypical and non-atypical hyperplasia patients, are required to identify women at risk of concurrent and future endometrial cancer, in whom preventative interventions can be targeted. Additional robust evidence is necessary to reliably inform treatment decisions for endometrial hyperplasia patients, something which is particularly pertinent for women who do not wish to undergo hysterectomy because of fertility issues or in whom hysterectomy is contraindicated because of comorbidities.

## Supporting information

**S1 Checklist. PRISMA 2009 checklist.**
(DOC)

**S1 Table. Characteristics of studies which assessed endometrial cancer in women with endometrial hyperplasia but unclear if cancer assessed within or after 3 months of hyperplasia diagnosis (n = 21).** EH- endometrial hyperplasia, EC- endometrial cancer, AH- atypical hyperplasia, SH-simple hyperplasia, CAH- complex atypical hyperplasia, D&C- dilation and curettage, SAH- simple atypical hyperplasia, CH-complex hyperplasia, TVUS- transvaginal ultrasound scan, NAH- non-atypical hyperplasia. [a]Mean age includes 8 patients diagnosed with endometrial cancer. [b]Likely includes n = 17 AH patients included in Agostini (2003) study. [c] Endometrial cancer rate per 1000 person-years was 9.3. [d] Follow-up time is for 249 hyperplasia patients who did not undergo hysterectomy initially.
(DOCX)

**S2 Table. Characteristics of studies which assessed the prevalence of concurrent endometrial cancer in women with endometrial hyperplasia which did not report time between biopsy and hysterectomy (n = 23).** EH- endometrial hyperplasia, EC- endometrial cancer, AH- atypical hyperplasia, SH-simple hyperplasia, CAH- complex atypical hyperplasia, D&C- dilation and curettage, SAH- simple atypical hyperplasia, CH-complex hyperplasia, NAH-non-atypical hyperplasia.
(DOCX)

**S3 Table. Summary of sub-group and sensitivity analyses for concurrent and future risk endometrial cancer in women with endometrial hyperplasia.** [a]Premenopausal women. [b] Postmenopausal women. [c]LNG-IUS (levonorgestrel intrauterine system) treated group. [d] Oral progesterone-treated group.
(DOCX)

**S1 Appendix. Search strategy used to identify relevant studies.**
(DOCX)

**S1 Fig. Funnel plot of studies investigating concurrent endometrial cancer diagnosed within three months of endometrial hyperplasia diagnosis.**
(TIF)

**S2 Fig. Funnel plot of studies investigating future endometrial cancer diagnosed after three months of endometrial hyperplasia diagnosis.**
(TIF)

**S3 Fig. Forest plot of incidence rates of endometrial cancer diagnosed after three months of endometrial hyperplasia diagnosis in studies with less than 24 months average follow-up.** [a]Premenopausal women.
(TIF)

**S4 Fig. Forest plot of incidence rates of endometrial cancer diagnosed after three months of endometrial hyperplasia diagnosis in studies with more than or equal to 24 months average follow-up.** [b]Postmenopausal women. [c]LNG-IUS (levonorgestrel intrauterine system) treated group. [d]Oral progesterone-treated group.
(TIF)

## Author Contributions

**Conceptualization:** Helen G. Coleman.

**Data curation:** Michelle T. Doherty, Omolara B. Sanni, Helen G. Coleman, Úna C. McMenamin.

**Formal analysis:** Omolara B. Sanni, Chris R. Cardwell, Úna C. McMenamin.

**Funding acquisition:** Helen G. Coleman.

**Investigation:** Michelle T. Doherty, Omolara B. Sanni, Helen G. Coleman, W. Glenn McCluggage, Declan Quinn, James Wylie, Úna C. McMenamin.

**Visualization:** Úna C. McMenamin.

**Writing – original draft:** Omolara B. Sanni, Úna C. McMenamin.

**Writing – review & editing:** Michelle T. Doherty, Omolara B. Sanni, Helen G. Coleman, Chris R. Cardwell, W. Glenn McCluggage, Declan Quinn, James Wylie, Úna C. McMenamin.

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
