## [Decision Letter · Decision Letter 0]

28 Jan 2020

PONE-D-19-35122

Concurrent and Future risk of Endometrial Cancer in women with Endometrial Hyperplasia: A systematic review and meta-analysis

PLOS ONE

Dear Dr McMenamin,

Thank you for submitting your manuscript to PLOS ONE. After careful consideration, we feel that it has merit but does not fully meet PLOS ONE’s publication criteria as it currently stands. Therefore, we invite you to submit a revised version of the manuscript that addresses the points raised during the review process.

While it is an imporatnt topic but I am not convinced if this study, by addition of a meta analysis that was not very helpfuldue due to the high heterogeneity of the studies, is superior to another review recently completed in 2019 covering very similar material. I would like to see more clarification of this hetergenity plus having results of sensitivity analysis proposed by reviewers.

We would appreciate receiving your revised manuscript by Mar 13 2020 11:59PM. To enhance the reproducibility of your results, we recommend that if applicable you deposit your laboratory protocols in protocols.io, where a protocol can be assigned its own identifier (DOI) such that it can be cited independently in the future. For instructions see: http://journals.plos.org/plosone/s/submission-guidelines#loc-laboratory-protocols

We look forward to receiving your revised manuscript.

Kind regards,

Omid Beiki, M.D., Ph.D.

Academic Editor

PLOS ONE

Journal Requirements:

3. Please provide any updates you might have since the original search was performed in September 2018, or please provide the rationale for ending your search at that time.

4. In the Methods, please specify any assessment of risk of bias that may affect the cumulative evidence (e.g., publication bias, selective reporting within studies). Please ensure that the specific method of assessment (funnel plot, Egger's test, Begg's test, etc) is mentioned.

Reviewers' comments:

Reviewer's Responses to Questions

**Comments to the Author**

1. Is the manuscript technically sound, and do the data support the conclusions?

Reviewer #1: Yes

Reviewer #2: Yes

Reviewer #3: Yes

2. Has the statistical analysis been performed appropriately and rigorously? 

Reviewer #1: Yes

Reviewer #2: Yes

Reviewer #3: Yes

3. Have the authors made all data underlying the findings in their manuscript fully available?

Reviewer #1: Yes

Reviewer #2: Yes

Reviewer #3: Yes

4. Is the manuscript presented in an intelligible fashion and written in standard English?

Reviewer #1: Yes

Reviewer #2: Yes

Reviewer #3: Yes

5. Review Comments to the Author

Reviewer #1: I have read with interest the manuscript of Dr. Doherty and colleagues. They are to be commended on a thorough and painstaking review of a relevant clinical question. The manuscript merits publication in my estimation, but there are few issues which I think should be addressed prior to acceptance. I have listed these below as either major (thematic) or minor (semantic).

Major

1. The stated goal of the manuscript is to more accurately assess the risk of concurrent or subsequent cancer among patients with hyperplastic conditions. While this is addressed in many ways one confounding variable I think should be addressed is the effect of treatment on observed concurrence of progression. In multiple of the studies patients we treated with IUD or oral progestin – and then went on to a more definitive procedure – however the indications for the procedure may impact the incidence/prevalence of a cancer diagnosis (ie a medically treated patient who ultimately required a hysterectomy may represent an “at risk” population) This should be discussed further.

2. Be careful about repeating (or listing) results in the discussion section. Reserve the discussion for the interpretation of the results. Much of page 20 can be moved to results, though discussion of how your data compares to the current literature should be in the discussion.

Minor

1. Table 1 – I would omit columns for which all of the data is the same (eg method of follow-up examination) – this will declutter the table and can be mentioned in the body of the tex.

2. Table 1 and 2 - There are abbreviations described that I did not see used in the table (eg TVUS); likewise there are abbreviations which are not standard and not described (Perma, and btw?)

3. Table 2 – keep the footnote references in the same column if possible e is in column 1 and the rest are in later columns (5-7)

4. Table 3 – I am not sure why the wheeler reference has two age groups – but only one results data point (we know that postmenopausal groups are older than premenopausal groups – I assume the progression rates are a summary stat – the ages should be summary in that case.

5. Discussion (page 21) I think a better descriptions of why a pathologist is prone to under-calling the pathology (as opposed to reluctant- as this implies he/she could but isn’t). Specifically the absence of invasion or stroma on slide makes the diagnosis more challenging and the guidelines generally recommend under- rather than over- calling on a biopsy

Reviewer #2: This manuscript describes a systematic review and meta-analysis evaluating the prevalence of endometrial cancer concurrent with an endometrial hyperplasia diagnosis and the future risk of endometrial cancer in women with such a diagnosis. Overall, the systematic review appears to have been done in accordance with standard guidelines and the data are generally clearly presented. I have only a few minor comments:

1. Page 4, line 80: The clinical guidelines are not stated exactly correctly. The cited guidelines state that endometrial biopsies should be done every 3 months until 2 consecutive negative biopsies are obtained.

2. Page 4, line 84: It should be previous studies evaluating the prevalence (not incidence) of concurrent endometrial cancer.

3. Page 7, line 152: The sensitivity analyses by quality score are not described accurately (less than or more than five would exclude studies with a score of 5).

4. Page 8, line 179: It would be helpful to add in a sentence summarizing the quality scores of the studies.

5. Page 8, line 192: The data reported are from Figure 2, not Figure 1.

6. When examining the results for the incidence of endometrial cancer (Figure 3), it appears that the highest estimates tend to be higher in studies that have a shorter duration of follow-up. A sensitivity analysis stratifying by duration of follow-up (perhaps <=24 months vs >24 months) may provide useful information about the timeframe after diagnosis of endometrial hyperplasia when the risk of progression is higher, which in turn may inform screening recommendations.

Reviewer #3: Concurrent and Future risk of Endometrial Cancer in women with Endometrial Hyperplasia: A systematic review and meta-analysis is an important area of research. However, there were numerous limitations. The empirical analysis was troublesome given the heterogeneity of findings, lack of population based studies, very low number of patients in some included studies, and lack of other characteristics that may influence the findings, such as pathology review, health care system, etc. To some extent, this feels more relevant as a systematic review (both another may have done this---Travaglino, 2019).

Intro was very well written

Line 177: “seven in North America, one in USA” Does in NA mean Canada? If so, please state.

By Line 182, I begin to wonder about the eligibility criteria and/or aim of each study included. In Table 1, it is hard to believe that only 42 cases of endometrial hyperplasia was found in the presumably chart review over a 10 year period (Osaka Medical Center) or 18 cases of endometrial hyperplasia over a 7 year period in Hospital la conception. There were several studies in which the number of cases was very small over many years which seems odd. That left me wondering about the aim of the study and if these included had a significant bias.

It seems that it would be reasonable in this electronic era to email authors of studies to determine the recruitment period as well as any other missing information populating the tables for their paper since a few studies had missing info.

Lines 214-218: this was unsatisfying results given the range was very wide (4.3to 287.9 per 1000 person-years with follow-up time ranging from three months to 23 years). Is there another way to analyze these data to have a more meaningful result? It seems that the meta-analyses was done by using data extracted from the studies versus obtaining the data from each site? Although it would take a lot of work, a much more meaningful analysis would use the raw data.

Line 220: I don’t know what “”Pooled analysis of five studies (including six estimates)” means. What does the six estimates mean. In this case it seems you are using raw data? But what is the estimates. Did I miss something?

Line 223: Do you think the results are sufficient to advance our understanding given the very high heterogeneity of the studies?

No line numbers in discussion but the stated limitation of prior studies ’ Moreover, most prior studies were conducted in single-center and tertiary referral centers which could overestimate risk in comparison to population-based studies.” Is also an issue here since I believe only one study was population based.

“An additional confounding factor is that in many institutions, most endometrial biopsies are reported by non-specialist pathologists” Having this information for this paper would be helpful although it would likely take work to find that out.

Overall, it seems that raw data were not used for the meta-analysis and this is a significant limitation.

6. PLOS authors have the option to publish the peer review history of their article (what does this mean?). If published, this will include your full peer review and any attached files.

Reviewer #1: No

Reviewer #2: No

Reviewer #3: No

---

## [Author Response · Author response to Decision Letter 0]

13 Mar 2020

Dear Prof Beiki, 

We thank the PLOS One reviewers and the editorial staff for taking the time to consider our revised manuscript and for their comments. We have responded to each comment below. We have conducted additional analyses as suggested (e.g. sensitivity analysis of duration of follow-up) and we have made numerous changes to the manuscript.

Please note that during the conduct of the suggested revisions, we noted a small error in the method used to pool the studies of concurrent endometrial cancer. We have now re-ran these analyses and although the results were largely unchanged, we have amended the tables and figures accordingly, as well as including a reference for the specific method used.

We think the revised manuscript is an improvement and hope that it is now suitable for publication in the PLOS One, but we are happy to consider further changes if necessary. 

Best wishes,

Dr Úna McMenamin (on behalf of the authors)

Journal Requirements:

*The manuscript has now been modified according to the style requirements of PLOS One.

*Captions for Supporting Information files have now been added to the end of our manuscript and in-text citations have been updated.

3. Please provide any updates you might have since the original search was performed in September 2018, or please provide the rationale for ending your search at that time.

*We have carried out an in-depth online literature review (e.g. Pubmed, Google Scholar) since the cut-off for our original search and did not find any articles that would have met our inclusion criteria. Therefore, we are confident that there are no subsequently published studies that would be relevant for inclusion in our review. Unfortunately, the time required to update the search strategy in all bibliographic databases and to independently review titles, abstracts and full-texts would exceed the time required to return the revised manuscript.

4. In the Methods, please specify any assessment of risk of bias that may affect the cumulative evidence (e.g., publication bias, selective reporting within studies). Please ensure that the specific method of assessment (funnel plot, Egger's test, Begg's test, etc) is mentioned.

*Funnel plots evaluating potential publication bias have now been added as supplementary figures (S1 and S2 Figs). There was no strong evidence of publication bias in either studies investigating concurrent endometrial cancer or progression to endometrial cancer and the methods and results text have been amended accordingly.

*The orcid ID for the corresponding author is now included within the Editorial Manager.

5. Review Comments to the Author

Reviewer #1: I have read with interest the manuscript of Dr. Doherty and colleagues. They are to be commended on a thorough and painstaking review of a relevant clinical question. The manuscript merits publication in my estimation, but there are few issues which I think should be addressed prior to acceptance. I have listed these below as either major (thematic) or minor (semantic).

*We thank the reviewer for their detailed review of our manuscript and we provide responses below to each of their queries.

Major

1. The stated goal of the manuscript is to more accurately assess the risk of concurrent or subsequent cancer among patients with hyperplastic conditions. While this is addressed in many ways one confounding variable I think should be addressed is the effect of treatment on observed concurrence of progression. In multiple of the studies patients we treated with IUD or oral progestin – and then went on to a more definitive procedure – however the indications for the procedure may impact the incidence/prevalence of a cancer diagnosis (ie a medically treated patient who ultimately required a hysterectomy may represent an “at risk” population) This should be discussed further.

*We agree with the reviewer and have now extended the limitations section of our discussion in an effort to acknowledge this and highlight the need for additional studies that include detailed information on progestin-based treatments, see below.

“…Moreover, few studies stratified results on progestogen use, which limited conclusions around the impact of progestin-based therapies on cancer risk and the lack of information on these treatments may have affected prevalence and incidence estimates. Considering the rising incidence of endometrial cancer in both pre and postmenopausal women61, temporal population-based estimates of progression to cancer in endometrial hyperplasia are warranted and such studies should account for treatments and other important clinical factors.”

61. Crosbie E, Morrison J. The emerging epidemic of endometrial cancer: Time to take action. Cochrane database Syst Rev. 2014;12:ED000095.

2. Be careful about repeating (or listing) results in the discussion section. Reserve the discussion for the interpretation of the results. Much of page 20 can be moved to results, though discussion of how your data compares to the current literature should be in the discussion.

*We have now reduced and removed some of the description of results from the discussion and have placed this in the results section. 

Minor

1. Table 1 – I would omit columns for which all of the data is the same (eg method of follow-up examination) – this will declutter the table and can be mentioned in the body of the tex.

*We have removed the column “Method of follow-up investigation” in Table 1 as suggested and described this in the body of the text.

2. Table 1 and 2 - There are abbreviations described that I did not see used in the table (eg TVUS); likewise there are abbreviations which are not standard and not described (Perma, and btw?)

*Only abbreviations used within the tables are now included in the table footnotes. 

3. Table 2 – keep the footnote references in the same column if possible e is in column 1 and the rest are in later columns (5-7)

*Footnote ‘e’ has been moved to column 5 to align with the other footnotes. We are keen to keep other footnotes in the current format, as we believe they better identify the stratified groups (columns 5-8; e.g. pre-, postmenopausal women) however we will modify this at the discretion of the Editor.

4. Table 3 – I am not sure why the wheeler reference has two age groups – but only one results data point (we know that postmenopausal groups are older than premenopausal groups – I assume the progression rates are a summary stat – the ages should be summary in that case.

*In Table 3, in an effort to avoid presenting missing information, we presented the average age of participants according to menopausal status for the study conducted by Wheeler et al. as unfortunately the authors did not present the mean age for the whole study population. 

5. Discussion (page 21) I think a better descriptions of why a pathologist is prone to under-calling the pathology (as opposed to reluctant- as this implies he/she could but isn’t). Specifically the absence of invasion or stroma on slide makes the diagnosis more challenging and the guidelines generally recommend under- rather than over- calling on a biopsy

*We have changed the wording of this sentence, as suggested by the reviewer to better reflect the problem of under-diagnosis of endometrial cancer in women with endometrial hyperplasia, please see below.

“Many endometrial biopsies, especially out-patient pipelle biopsies and curettages, are scanty with little tissue represented and the absence of invasion or stroma on slides makes the diagnosis more challenging and risks potentially missing the most significant lesion. Therefore, pathologists may be less likely to make a firm cancer diagnosis.”

Reviewer #2: This manuscript describes a systematic review and meta-analysis evaluating the prevalence of endometrial cancer concurrent with an endometrial hyperplasia diagnosis and the future risk of endometrial cancer in women with such a diagnosis. Overall, the systematic review appears to have been done in accordance with standard guidelines and the data are generally clearly presented. I have only a few minor comments:

*We thank the reviewer for their positive comments on our manuscript and provide responses to their specific questions below.

1. Page 4, line 80: The clinical guidelines are not stated exactly correctly. The cited guidelines state that endometrial biopsies should be done every 3 months until 2 consecutive negative biopsies are obtained.

*We have modified the sentence as requested, to better clarify with the current clinical guidelines.

2. Page 4, line 84: It should be previous studies evaluating the prevalence (not incidence) of concurrent endometrial cancer.

*We have corrected the sentence accordingly.

3. Page 7, line 152: The sensitivity analyses by quality score are not described accurately (less than or more than five would exclude studies with a score of 5).

*We have now changed this line to “…restricting to studies with a quality score of less than five and more than or equal to five.”

4. Page 8, line 179: It would be helpful to add in a sentence summarizing the quality scores of the studies.

*In the description of study characteristics, we now state that “most studies were classified as being of low-to-moderate quality, with quality scores ranging from four to six out of a total of nine.” 

5. Page 8, line 192: The data reported are from Figure 2, not Figure 1.

*Thank you, we have now rectified this error.

6. When examining the results for the incidence of endometrial cancer (Figure 3), it appears that the highest estimates tend to be higher in studies that have a shorter duration of follow-up. A sensitivity analysis stratifying by duration of follow-up (perhaps <=24 months vs >24 months) may provide useful information about the timeframe after diagnosis of endometrial hyperplasia when the risk of progression is higher, which in turn may inform screening recommendations.

*We agree that a sensitivity analysis based on average duration of follow-up would be useful and have included this as supplemental material (Supplemental Figures 1 and 2). The risk of progressing to endometrial cancer was indeed higher in studies with shorter follow-up; pooled analysis of three studies with an average follow-up of less than 24 months (including 121 women with atypical hyperplasia), showed that the incidence rate of endometrial cancer was 107 per 1,000 person-years (95% CI 27.6, 414.9), equivalent to 10.7% per year, with high heterogeneity observed (I2=78.3%). Among three studies with an average follow-up of less than or equal to 24 months, pooled analysis of 67 women with atypical hyperplasia showed a lower incidence rate of endometrial cancer (66.1 per 1,000 person-years, 95% CI 30.4, 144), equivalent to 6.6% per year, with moderate heterogeneity observed (I2=49.5%). Only one study evaluated progression to endometrial cancer specifically in women with non-atypical endometrial hyperplasia, which therefore precluded meta-analysis.

Four studies that included both atypical and non-atypical hyperplasia patients, had an average follow-up of more than or equal to 24 months (n=973 women) and pooled analysis showed that the incidence rate of endometrial cancer was 9.2 per 1,000 person-years (95% CI 5.4, 15.9), equivalent to 0.9% per year, with high heterogeneity observed (I2=87.7%). We identified no studies that investigated both atypical and non-atypical hyperplasia patients with an average follow-up of less than 24 months.

We have amended the methods, results and discussion section accordingly to describe and interpret these additional sensitivity analyses.

Reviewer #3: Concurrent and Future risk of Endometrial Cancer in women with Endometrial Hyperplasia: A systematic review and meta-analysis is an important area of research. However, there were numerous limitations. The empirical analysis was troublesome given the heterogeneity of findings, lack of population based studies, very low number of patients in some included studies, and lack of other characteristics that may influence the findings, such as pathology review, health care system, etc. To some extent, this feels more relevant as a systematic review (both another may have done this---Travaglino, 2019).

*We thank the reviewer for conducting a thorough review of our manuscript. As described earlier, we have highlighted these specific limitations within the discussion of our manuscript but we have now added additional recommendations for the conduct of future studies that are population-based, include both pre and postmenopausal women and incorporate important clinical data such as progestogen therapies. In an effort to avoid the inclusion of very small studies from the outset, our pre-specified article selection criteria excluded studies with fewer than 10 cases of endometrial hyperplasia. Moreover, we conducted sensitivity analysis restricting to higher quality studies, classified using a validated quality assessment tool.

We also now include a description of another recently conducted systematic review by Travaglino et al1, which investigated risk of occult endometrial cancer in complex non‑atypical endometrial hyperplasia. However, unlike our systematic review, there was no time cut-off used to define concurrent endometrial cancer; the time from endometrial hyperplasia diagnosis to hysterectomy ranged from 4 days to 7 years within individual studies meaning that outcomes may have represented progression to cancer rather than coexistent cancer. Moreover, the review by Travaglino et al. restricted to women diagnosed with complex non‑atypical endometrial hyperplasia whereas our review includes all types of hyperplasia, thereby adding considerable value to the current evidence base.

1Travaglino A, Raffone A, Saccone G, et al. Significant risk of occult cancer in complex non-atypical endometrial hyperplasia. Arch Gynecol Obstet 2019; 300: 1147-1154.

Intro was very well written

*We thank the reviewer for their positive feedback.

Line 177: “seven in North America, one in USA” Does in NA mean Canada? If so, please state.

*We have now revised this sentence which we hope clarifies the description of study location; “Fifteen studies were conducted in Europe, 12 in Asia, eight in North America (one study included patients from the US and Europe) and one in Australia”

By Line 182, I begin to wonder about the eligibility criteria and/or aim of each study included. In Table 1, it is hard to believe that only 42 cases of endometrial hyperplasia was found in the presumably chart review over a 10 year period (Osaka Medical Center) or 18 cases of endometrial hyperplasia over a 7 year period in Hospital la conception. There were several studies in which the number of cases was very small over many years which seems odd. That left me wondering about the aim of the study and if these included had a significant bias.

* We were restricted to the information that authors provided in the original manuscripts but we have no reason to believe that this information is incorrect. Both studies that the reviewer highlights were conducted in single-centre institutions and only included women diagnosed with atypical hyperplasia, the less common type of endometrial hyperplasia. The aim of the study by Agostini et al (2003), conducted within the Hospital la Conception, was to determine the rate of concurrent endometrial cancer in women diagnosed with atypical hyperplasia using hysteroscopic resection, so therefore atypical hyperplasia cases diagnosed using other methods may not have been included. We have modified the information in Table 1 and in the results section to better clarify the method of hyperplasia diagnosis within included studies.

Within our discussion, we acknowledge that small case numbers within individual studies is a limitation of our review and we have called for larger (population-based) studies to be carried out. As described, in an effort to minimise the impact from these studies, we conducted sensitivity analysis restricting to higher quality studies and our pre-specified article selection criteria excluded studies with fewer than 10 cases of endometrial hyperplasia. This therefore helped ensure the inclusion of meaningful estimates of concurrent cancer risk or progression to cancer.

It seems that it would be reasonable in this electronic era to email authors of studies to determine the recruitment period as well as any other missing information populating the tables for their paper since a few studies had missing info.

*During the conduct of the search strategy, efforts were made to email some authors for additional information; however, as there was limited response, we were restricted to including the information presented in the study articles. 

Lines 214-218: this was unsatisfying results given the range was very wide (4.3to 287.9 per 1000 person-years with follow-up time ranging from three months to 23 years). Is there another way to analyze these data to have a more meaningful result? It seems that the meta-analyses was done by using data extracted from the studies versus obtaining the data from each site? Although it would take a lot of work, a much more meaningful analysis would use the raw data.

*As the reviewer highlights, obtaining the raw data from individual studies included in this systematic review would take a considerable amount of time and unfortunately would not be feasible within the time requested to return the manuscript (six weeks).

Line 220: I don’t know what “”Pooled analysis of five studies (including six estimates)” means. What does the six estimates mean. In this case it seems you are using raw data? But what is the estimates. Did I miss something?

*In this particular analysis, one study only presented estimates according to menopausal status (Brownfoot et al. 2014) and so we included both of these estimates (pre- and postmenopausal status), as outline in Figure 3. We have modified this line to clarify that two estimates were extracted from one individual study.

Line 223: Do you think the results are sufficient to advance our understanding given the very high heterogeneity of the studies?

*We agree that the high heterogeneity observed in our meta-analysis is a limitation and we have acknowledged this in our discussion. We also suggest that the observed heterogeneity could be due to differences in study populations, methods used to diagnose endometrial hyperplasia, as well as the wide variation in the time between biopsy and hysterectomy, which ranged from three months to 12 years in studies of endometrial cancer progression. Despite this, the findings from our comprehensive systematic review, which used clinically relevant time cut-off points for the identification of cancer, provide a broad overview of the current evidence to date on the risk of concurrent and future risk of endometrial cancer in women diagnosed with endometrial hyperplasia and will therefore be of value to clinicians, researchers and patients. As described earlier, we now include a sensitivity analysis stratifying on average duration of follow-up, which resulted in reduced heterogeneity (e.g. 49.5% heterogeneity in pooled analysis of progression to endometrial cancer in women with atypical hyperplasia among studies with less than 24 months average follow-up). 

Overall, we believe that our systematic review will stimulate the conduct of higher quality, population-representative studies, which will allow for the calculation of more accurate risk estimates to better-inform decision making between endometrial hyperplasia patients and their clinicians, particularly with respect to surveillance and treatments options.

No line numbers in discussion but the stated limitation of prior studies ’ Moreover, most prior studies were conducted in single-center and tertiary referral centers which could overestimate risk in comparison to population-based studies.” Is also an issue here since I believe only one study was population based.

*Within our discussion, we already call for the conduct of future population-based studies in order to provide more accurate estimates of risk of progression to endometrial cancer in women diagnosed with endometrial hyperplasia. As outlined earlier however, we have now extended this to request that population-based studies also ensure the inclusion of important clinical data to allow stratification by factors that may affect risk of endometrial cancer. 

We have also now included line numbers in the discussion.

“An additional confounding factor is that in many institutions, most endometrial biopsies are reported by non-specialist pathologists” Having this information for this paper would be helpful although it would likely take work to find that out.

*Unfortunately, information on the type of pathologist conducting histopathological review of biopsies was not uniformly reported within the individual studies. Moreover, details of the reporting pathologists was not part of our pre-specified data extraction plan for this systematic review.

Overall, it seems that raw data were not used for the meta-analysis and this is a significant limitation.

*Please see earlier response with respect to the lack of availability of raw data for this systematic review.

---

## [Decision Letter · Decision Letter 1]

10 Apr 2020

Concurrent and future risk of endometrial cancer in women with endometrial hyperplasia: a systematic review and meta-analysis

PONE-D-19-35122R1

Dear Dr. McMenamin,

We are pleased to inform you that your manuscript has been judged scientifically suitable for publication and will be formally accepted for publication once it complies with all outstanding technical requirements.

With kind regards,

Omid Beiki, M.D., Ph.D.

Academic Editor

PLOS ONE

Additional Editor Comments (optional):

Reviewers' comments:

Reviewer's Responses to Questions

**Comments to the Author**

1. If the authors have adequately addressed your comments raised in a previous round of review and you feel that this manuscript is now acceptable for publication, you may indicate that here to bypass the “Comments to the Author” section, enter your conflict of interest statement in the “Confidential to Editor” section, and submit your "Accept" recommendation.

Reviewer #1: All comments have been addressed

Reviewer #2: All comments have been addressed

Reviewer #3: All comments have been addressed

2. Is the manuscript technically sound, and do the data support the conclusions?

Reviewer #1: Yes

Reviewer #2: (No Response)

Reviewer #3: Yes

3. Has the statistical analysis been performed appropriately and rigorously? 

Reviewer #1: Yes

Reviewer #2: (No Response)

Reviewer #3: Yes

4. Have the authors made all data underlying the findings in their manuscript fully available?

Reviewer #1: Yes

Reviewer #2: (No Response)

Reviewer #3: Yes

5. Is the manuscript presented in an intelligible fashion and written in standard English?

Reviewer #1: Yes

Reviewer #2: (No Response)

Reviewer #3: Yes

6. Review Comments to the Author

Reviewer #1: The revisions appear appropriate. s

Reviewer #2: (No Response)

Reviewer #3: Thanks for your thorough consideration of the tough questions. Although the issue of missing data and small number of cases is a serious limitation, I agree that this might help others gather more complete data for analyses.

7. PLOS authors have the option to publish the peer review history of their article (what does this mean?). If published, this will include your full peer review and any attached files.

Reviewer #1: No

Reviewer #2: No

Reviewer #3: No

---

## [Editor Report · Acceptance letter]

16 Apr 2020

PONE-D-19-35122R1 

Concurrent and future risk of endometrial cancer in women with endometrial hyperplasia: a systematic review and meta-analysis 

Dear Dr. McMenamin:

I am pleased to inform you that your manuscript has been deemed suitable for publication in PLOS ONE. Congratulations! Your manuscript is now with our production department. 

With kind regards,

on behalf of

Dr. Omid Beiki 

Academic Editor

PLOS ONE